# Dynamic Prompt Generation for Interactive 3D Medical Image Segmentation Training

Tidiane Camaret Ndir[1,2][0009−0009−9523−2157], Alexander
Pfefferle[2,3][0009−0003−5457−7526], and Robin T.
Schirrmeister[1][0000−0002−5518−7445]

[1] Medical Physics, Department of Diagnostic and Interventional Radiology, Medical
Center—University of Freiburg, Faculty of Medicine, University of Freiburg, Freiburg,
Germany
[2] University of Freiburg, Freiburg, Germany
[3] ELLIS Institute Tübingen, Tübingen, Germany
{tidiane.camaret.ndir}@uniklinik-freiburg.de

**Abstract.** Interactive 3D biomedical image segmentation requires effi-
cient models that can iteratively refine predictions based on user prompts.
Current foundation models either lack volumetric awareness or suffer
from limited interactive capabilities. We propose a training strategy that
combines dynamic volumetric prompt generation with content-aware adap-
tive cropping to optimize the use of the image encoder. Our method sim-
ulates realistic user interaction patterns during training while addressing
the computational challenges of learning from sequential refinement feed-
back on a single GPU. For efficient training, we initialize our network
using the publicly available weights from the nnInteractive segmentation
model. Evaluation on the **Foundation Models for Interactive 3D
Biomedical Image Segmentation** competition demonstrates strong
performance with an average final Dice score of 0.6385, normalized sur-
face distance of 0.6614, and area-under-the-curve metrics of 2.4799 (Dice)
and 2.5671 (NSD).

**Keywords:** Interactive segmentation · 3D medical imaging · Dynamic
prompt generation

## 1    Introduction

### 1.1    Background

Accurate 3D medical image segmentation is essential for clinical and research
applications. The increasing volume and complexity of biomedical data sets re-
quire robust interactive segmentation methods that efficiently incorporate user
feedback.

The **Foundation Models for Interactive 3D Biomedical Image Seg-
mentation** (SEGFM3D) challenge evaluates interactive segmentation through
progressive refinement: models receive an initial bounding box, which may or

may not be provided depending on the case, followed by 1-5 iterative click refinements to correct errors. Models must complete segmentation within 90 seconds per class, enforcing computational efficiency. The challenge requires optimal utilization of heterogeneous user prompts, bounding boxes for localization and clicks for refinement, across diverse 3D anatomical structures and imaging modalities.

## 1.2   Related Work

Foundation models for image segmentation, particularly SAM [5] and SAM2 [11], have demonstrated strong zero-shot performance on natural images. Medical adaptations including MedSAM [7] and MedSAM2 [9] fine-tune these models on medical datasets but exhibit limited interactive refinement capabilities, particularly for iterative correction workflows.

Recent advances in interactive 3D medical segmentation include SegVol [1], SAM-Med3D [13], VISTA3D [3], and nnInteractive [2]. These methods explore different approaches to incorporating user feedback in volumetric medical image analysis, establishing the foundation for interactive refinement in clinical workflows.

However, those methods do not optimally utilize all available information in the SEGFM3D challenge: initial bounding boxes, previous iteration segmentations, and competition-generated clicks.

Existing approaches exhibit fundamental limitations: VISTA3D [3] supports only click prompts, lacking bounding box and iterative refinement capabilities. SegVol [1] accepts only bounding boxes without click-based correction. SAM-Med3D [13] operates at low resolution, insufficient for accurate 3D segmentation. nnInteractive [2] accepts multiple prompts but restricts bounding boxes to 2D slices, ignoring 3D spatial context.

A critical challenge is the circular dependency in training interactive models: generating realistic training signals requires iterative predictions and error-based click simulation, yet the model must learn from these self-generated interactions.

## 1.3   Solution and Contribution

We present two key contributions to address the limitations of current interactive 3D segmentation methods:

First, we introduce a dynamic prompt generation strategy that simulates realistic user interaction patterns during training. This approach generates clicks by identifying the largest connected error components between predictions and ground truth, mimicking the competition evaluation settings.

Second, we propose content-aware dynamic cropping that adapts the model field of view sizes based on anatomical context. This ensures complete capture of target structures while maintaining computational efficiency.

## 2   Method

### 2.1   General Network Architecture

We base our solution on the nnInteractive [2] method and adopt a 3D Residual Encoder U-Net [12] architecture, combining the spatial precision of U-Net with the gradient flow benefits of residual connections, specifically optimized for interactive volumetric medical image segmentation.

The encoder consists of six stages with progressively increasing feature dimensions $(32, 64, 128, 256, 320, 320)$ while reducing spatial resolution through $3 \times 3 \times 3$ convolutions with stride 2. Each encoder stage incorporates residual blocks following the ResNet paradigm:

$$\mathcal{F}_{\text{res}}(x) = x + \mathcal{B}(x), \tag{1}$$

where $x$ is the input feature map, $\mathcal{F}_{\text{res}}(x)$ denotes the residual block output and $\mathcal{B}(x)$ represents two consecutive 3D convolutional layers with GroupNorm and LeakyReLU activations.

The decoder mirrors the encoder structure with transposed convolutions for upsampling. We only compute the loss on the final segmentation prediction and do not employ deep supervision, which is sometimes used to get additional training signal using segmentation predictions obtained from earlier layers of the decoder. We found that unnecessary since we initialize our parameters from the pretrained nnInteractive segmentation model.

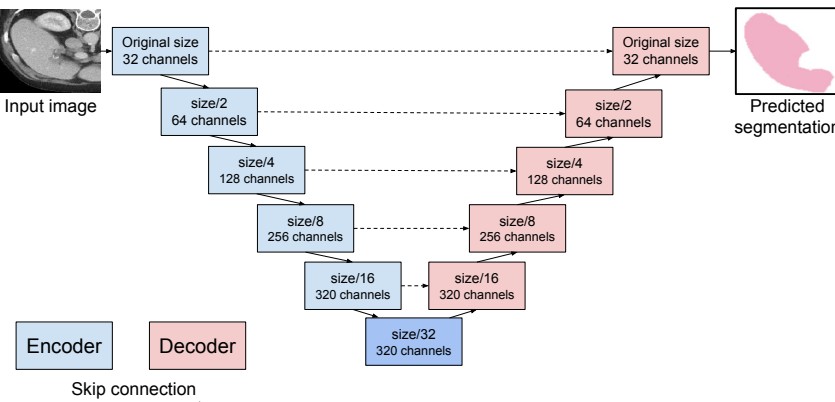

Fig. 1: 3D Residual Encoder U-Net Architecture : A U-Net with residual blocks processing 3D input through 6-stage encoder-decoder with skip connections.

## 2.2    Prompt Encoding

Following the nnInteractive [2] strategy, we encode heterogeneous prompts (bounding boxes, positive/negative clicks and previous segmentation masks) as additional input channels concatenated with the image data, as illustrated in Figure 2.

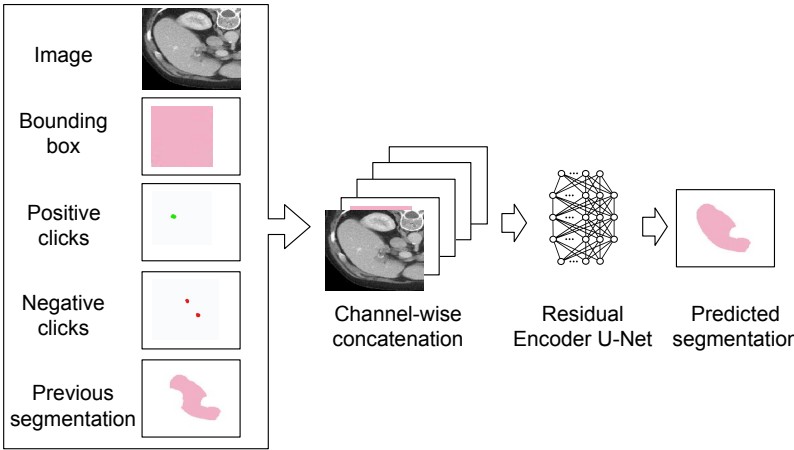

Fig. 2: Prompt encoding : Prompts are concatenated with the original image as extra channels.

**Multi-Modal Prompt Representation** Our input tensor $\mathcal{X} \in \mathbb{R}^{C \times D \times H \times W}$ encodes information in 5 channels:

1. **Image Channel** ($\mathcal{X}_0$): The normalized 3D medical image with intensity values in $[0, 255]$.
2. **Bounding Box Channel** ($\mathcal{X}_1$): A binary mask where voxels inside the initial bounding box are set to 1, and 0 elsewhere:

$$\mathcal{X}_1(i,j,k) = \begin{cases} 1 & \text{if } (i,j,k) \in \mathcal{B} \\ 0 & \text{otherwise,} \end{cases} \tag{2}$$

   where $\mathcal{B}$ represents the 3D bounding box region.
3. **Positive Click Channel** ($\mathcal{X}_2$): Encodes user-provided positive clicks indicating regions that should be included in the segmentation. Each click is represented as a sphere of radius 4 voxels:

$$\mathcal{X}_2(i,j,k) = \begin{cases} 1 & \text{if } \min_{c \in \mathcal{C}^+} \|(i,j,k) - c\| \leq 4 \\ 0 & \text{otherwise,} \end{cases} \tag{3}$$

where $\mathcal{C}^+$ is the set of positive click coordinates.

4. **Negative Click Channel** ($\mathcal{X}_3$): Similarly encodes negative clicks indicating regions to exclude:

$$\mathcal{X}_3(i,j,k) = \begin{cases} 1 & \text{if } \min_{c \in \mathcal{C}^-} \|(i,j,k) - c\| \leq 4 \\ 0 & \text{otherwise.} \end{cases} \tag{4}$$

5. **Previous Segmentation Channel** ($\mathcal{X}_4$): Contains the binary mask from the previous iteration, enabling the model to refine its predictions based on prior outputs. For the initial iteration, this channel is zero-initialized.

**Handling Missing Prompts** For anatomical structures without meaningful bounding boxes (e.g., vessels, myocardium), we generate a default bounding box covering the central third of the volume:

$$\mathcal{B}_{\text{default}} = \left[ \frac{D}{3}, \frac{2D}{3} \right] \times \left[ \frac{H}{3}, \frac{2H}{3} \right] \times \left[ \frac{W}{3}, \frac{2W}{3} \right]. \tag{5}$$

This heuristic ensures consistent input dimensionality while providing a reasonable initialization region for structures that typically appear near the image center.

### 2.3   Interaction Simulation

Training interactive segmentation models presents a unique challenge: the model must learn from a feedback that depends on its own predictions. We address this circular dependency by implementing a dynamic interaction simulation strategy that mimics the competition's click generation logic during training. We simulate the iterative refinement process through a two-stage approach.

1. **Interaction simulation stage (no gradient computation)** : We run the network with the original image and the bounding box and all other prompts channels set to zero to generate an initial prediction, identify errors via connected component analysis, and generate click prompts, as illustrated in Figure 3.
2. **Training Stage (with gradient computation)** : We perform a forward pass with all prompts (bounding box, generated clicks and previous segmentation) to produce the final prediction. Only this final prediction is used to compute the loss and update the network weights through backpropagation.

**Stochastic Interaction Sampling** To ensure robust and efficient training, we stochastically choose one of two settings for each batch during training: (1) No click and no given previous segmentation, or (2) a single click and the network's prediction from setting (1) as the given previous segmentation. In the setting with a given network segmentation (2), we transform the previous segmentation

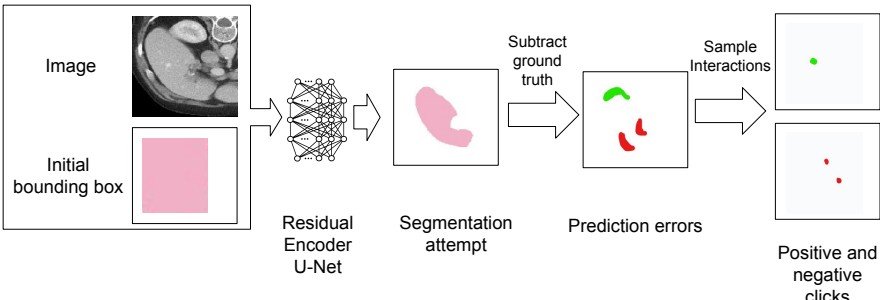

Fig. 3: Dynamic interaction simulation during training. The model generates an initial segmentation using the bounding box prompt without gradient computation. Click prompts are then generated based on error analysis between this prediction and ground truth. Note : The final segmentation is produced using all prompts, and only this final prediction is used to compute the loss and update network weights via backpropagation

into a hard binary segmentation and do not backpropagate through it to reduce GPU memory usage.

$$1\text{-click} \sim \text{Bernoulli}(p_{\text{click}}), \tag{6}$$

where $p_{\text{click}} = 0.5$. This distribution was chosen to have a simple training setup that can generalize well to the evaluation. Using the prediction of the network under training as the given prior segmentation should generalize well to evaluation where the same setting is used.

**Error-based Click Generation** For each sampled interaction level, we generate clicks by identifying and targeting segmentation errors:

1. **Error Detection**: We compute the binary error mask between the predicted segmentation $\hat{Y}$ and ground truth $Y$:

$$\mathcal{E} = (\hat{Y} > 0.5) \oplus (Y > 0.5), \tag{7}$$

where $\oplus$ denotes the XOR operation.
2. **Connected Component Analysis**: We identify spatially connected error regions using 26-connectivity in 3D:

$$\mathcal{C} = \text{CC3D}(\mathcal{E}, \text{connectivity} = 26). \tag{8}$$

3. **Largest Error Selection**: We target the largest connected error component to maximize correction impact:

$$c^* = \arg\max_{c \in \mathcal{C}} |\{(i, j, k) : \mathcal{C}(i, j, k) = c\}|. \tag{9}$$

4. **Click Center Determination**: We compute the Euclidean Distance Transform (EDT) of the largest error component and sample the click location from the maximum distance point:

$$\text{click\_center} = \arg\max_{(i,j,k) \in c^*} \text{EDT}(c^*). \tag{10}$$

5. **Click Type Assignment**: The click type (positive or negative) is determined by the ground truth value at the click location:

$$\text{click\_type} = \begin{cases} \text{positive} & \text{if } Y(\text{click\_center}) = 1 \\ \text{negative} & \text{if } Y(\text{click\_center}) = 0. \end{cases} \tag{11}$$

### 2.4 Warm Starting

We initialize our model with pre-trained weights from nnInteractive v1.0, which has demonstrated strong performance on interactive medical image segmentation tasks. However, the original nnInteractive architecture was designed to accommodate a larger set of interaction prompts beyond those used in the SEGFM3D challenge. To leverage the pre-trained weights while maintaining architectural compatibility, we fill the unused channels with zero tensors. Note also that nnInteractive was trained with 2d bounding boxes within the 3d volume, while we train using the full 3d bounding box to make full use of the volumetric information.

### 2.5 Processing Details

**Handling Large Images** To process large 3D medical images within memory constraints while preserving the spatial context around the target region, we implement a content-aware dynamic cropping strategy. Given an input patch size of $192 \times 192 \times 192$ and a bounding box $\mathcal{B}$, we:

1. **Compute the required zoom factor** to ensure the bounding box fits within the patch with a margin:

$$z = \max\left(\frac{\text{bbox\_size} + \text{patch\_size}/3}{\text{patch\_size}}, 1\right) \tag{12}$$

2. **Scale the patch size** to accommodate the entire bounding box:

$$\text{scaled\_patch\_size} = \lceil z \times \text{patch\_size} \rceil \tag{13}$$

3. **Center the crop** around the bounding box center and pad if necessary to handle boundary cases.
4. **Resize to target dimensions** during inference to maintain consistent input size for the network.

This approach ensures that: (i) the entire bounding box is captured regardless of its size, (ii) sufficient context (33% margin) is preserved around the target region, and (iii) the aspect ratio of anatomical structures is maintained during processing.

**Post-processing** We use the per-class sigmoid-transformed outputs of the networks as binary probabilities to create the multiclass segmentation map. First, we compute the probability of the background class as the probability that none of the foreground classes are present. This is calculated as the product of the negative probabilities of all other classes: $p_{bg} = \prod_{c=1}^{K}(1 - p_c)$, where $p_c$ is the probability of class $c$, and the product runs over all K foreground classes. The predicted class at each location is then assigned by taking the argmax over the background and all foreground probabilities: class $= \arg\max_{i \in \{0, \dots K\}} p_i$, where $p_0 = p_{bg}$ is the background probability.

**Inference Optimization** We followed the Docker optimizations in DAFT [10] and reduced the size and number of layers in the image by: (i) using a minimal base image (`python : 3.13-slim-bookworm`), (ii) disabling pip caching (`PIP_NO_CACHE_DIR=1`), and (iii) squashing layers with `docker-squash`.

### 2.6   Loss Function

We employ a compound loss function combining Dice loss and cross-entropy, as such hybrid approaches have proven robust across various medical image segmentation tasks [6]. The total loss is formulated as:

$$\mathcal{L}_{\text{seg}} = -\alpha \mathcal{L}_{\text{Dice}} + \mathcal{L}_{\text{CE}}, \tag{14}$$

where $\alpha = 10$.

**Dice Loss** The Dice loss promotes global consistency and handles class imbalance:

$$\mathcal{L}_{\text{Dice}} = 1 - \frac{2 \sum_i p_i g_i + \epsilon}{\sum_i p_i + \sum_i g_i + \epsilon}, \tag{15}$$

where $p_i \in [0, 1]$ represents the predicted probability at voxel $i$, $g_i \in \{0, 1\}$ is the ground truth label, and $\epsilon = 10^{-5}$ ensures numerical stability.

**Cross-Entropy Loss** The binary cross-entropy loss provides strong gradients for individual voxel classification:

$$\mathcal{L}_{\text{CE}} = -\frac{1}{N} \sum_{i=1}^{N} [g_i \log(p_i) + (1 - g_i) \log(1 - p_i)], \tag{16}$$

where $N$ is the total number of voxels.

## 3    Experiments

### 3.1    Dataset and evaluation metrics

The development set is an extension of the CVPR 2024 MedSAM on Laptop Challenge [8], including more 3D cases from public datasets[4] and covering commonly used 3D modalities, such as Computed Tomography (CT), Magnetic Resonance Imaging (MRI), Positron Emission Tomography (PET), Ultrasound, and Microscopy images. The hidden testing set is created by a community effort where all the cases are unpublished. The annotations are either provided by the data contributors or annotated by the challenge organizer with 3D Slicer [4] and MedSAM2 [9]. In addition to using all training cases, the challenge contains a coreset track, where participants can select 10% of the total training cases for model development.

For each iterative segmentation, the evaluation metrics include Dice Similarity Coefficient (DSC) and Normalized Surface Distance (NSD) to evaluate the segmentation region overlap and boundary distance, respectively. The final metrics used for the ranking are:

- DSC_AUC and NSD_AUC Scores: AUC (Area Under the Curve) for DSC and NSD is used to measure cumulative improvement with interactions. The AUC quantifies the cumulative performance improvement over the five click predictions, providing a holistic view of the segmentation refinement process. It is computed only over the click predictions without considering the initial bounding box prediction as it is optional.
- Final DSC and NSD Scores after all refinements, indicating the model's final segmentation performance.

In addition, the algorithm runtime will be limited to 90 seconds per class, measured by the total runtime of the submitted docker container. Exceeding this limit will lead to all DSC and NSD metrics being set to 0 for that test case. The ranking was computed for each metric and test case and then averaged to get the final ranking in the challenge.

### 3.2    Implementation details

**Preprocessing** Following the practice in MedSAM [7], all images were processed to npz format with an intensity range of $[0, 255]$. Specifically, for CT images, we initially normalized the Hounsfield units using typical window width and level values: soft tissues (W:400, L:40), lung (W:1500, L:-160), brain (W:80, L:40), and bone (W:1800, L:400). Subsequently, the intensity values were rescaled to the range of $[0, 255]$. For other images, we clipped the intensity values to the range between the 0.5th and 99.5th percentiles before rescaling them to the range of $[0, 255]$. If the original intensity range is already in $[0, 255]$, no preprocessing was applied.

---

[4] A complete list is available at https://medsam-datasetlist.github.io/

**Environment settings** The development environments and requirements are presented in Table 1.

Table 1: Development environments and requirements

| | |
|---|---|
| System | Ubuntu 20.04.6 LTS |
| CPU | AMD EPYC 7713P 64-Core Processor |
| RAM | 8×2GB; 2.67MT/s |
| GPU (number and type) | One NVIDIA RTX 6000 Ada Generation (50 G) |
| CUDA version | 12.3 |
| Programming language | Python 3.12.10 |
| Deep learning framework | torch 2.6.0 |

**Training Protocols** Our training configuration is summarized in Table 2.

    **Data Augmentation.** We employ minimal augmentation as extensive transformations have been shown to degrade performance on the evaluation task. Our pipeline consists of: (i) intensity normalization to [0, 255], (ii) random class selection when multiple objects are present, (iii) cropping to non-zero regions around the selected object, and (iv) resizing to the model's input patch size.

    **Data Sampling Strategy.** We train on 10% of the dataset with uniform sampling across all modalities. Each training iteration randomly selects one segmentation class per image to ensure balanced exposure to different anatomical structures.

Table 2: Training protocols

| | |
|---|---|
| Pre-trained weights | nnInteractive v1.0 |
| Batch size | 2 |
| Patch size | 192×192×192 |
| Total epochs | 25 |
| Optimizer | AdamW |
| Initial learning rate (lr) | 1e-4 |
| Lr decay schedule | constant learning rate |
| Training time | 16 hours |
| Loss function | - 10 * Dice Loss + Cross Entropy Loss |
| Number of model parameters | 102 M |
| Number of flops | 10000G |

# 4 Results and discussion

## 4.1 Quantitative results on validation set

Table 3: Quantitative evaluation results on the validation set for the **coreset track**

| Modality | Methods | DSC AUC | NSD AUC | DSC Final | NSD Final |
|---|---|---|---|---|---|
| CT | SAM-Med3D | 2.2408 | 2.2213 | 0.5590 | 0.5558 |
|  | VISTA3D | 3.1689 | 3.2652 | 0.8041 | 0.8344 |
|  | SegVol | 2.9809 | 3.1235 | 0.7452 | 0.7809 |
|  | nnInteractive | **3.4337** | **3.5743** | **0.8764** | **0.9165** |
|  | Ours | 3.2197 | 3.3272 | 0.8197 | 0.8511 |
| MRI | SAM-Med3D | 1.5222 | 1.5226 | 0.3903 | 0.3964 |
|  | VISTA3D | 2.5895 | 2.9683 | 0.6545 | 0.7493 |
|  | SegVol | 2.6719 | **3.1535** | 0.6680 | 0.7884 |
|  | nnInteractive | **2.6975** | 3.0292 | **0.7302** | **0.8227** |
|  | Ours | 1.5944 | 1.6136 | 0.4222 | 0.4254 |
| Microscopy | SAM-Med3D | 0.1163 | 0 | 0.0291 | 0 |
|  | VISTA3D | 2.1196 | 3.2259 | 0.5478 | 0.8243 |
|  | SegVol | 1.6846 | 2.9716 | 0.4211 | 0.7429 |
|  | nnInteractive | 2.3311 | 3.1109 | 0.5943 | 0.7890 |
|  | Ours | **2.9475** | **3.8109** | **0.7531** | **0.9606** |
| PET | SAM-Med3D | 2.1304 | 1.7250 | 0.5344 | 0.4560 |
|  | VISTA3D | 2.6398 | 2.3998 | 0.6779 | 0.6227 |
|  | SegVol | 2.9683 | 2.8563 | 0.7421 | 0.7141 |
|  | nnInteractive | 3.1877 | 3.0722 | 0.8156 | **0.7915** |
|  | Ours | **3.2278** | **3.0727** | **0.8186** | 0.7841 |
| Ultrasound | SAM-Med3D | 1.4347 | 1.9176 | 0.4102 | 0.5435 |
|  | VISTA3D | 2.8655 | 2.8441 | 0.8105 | 0.8079 |
|  | SegVol | 1.2438 | 1.8045 | 0.3109 | 0.4511 |
|  | nnInteractive | 3.3481 | 3.3236 | 0.8547 | 0.8494 |
|  | Ours | **3.5364** | **3.6196** | **0.8946** | **0.9165** |

Table 3 presents our performance on the validation set for the coreset track for the five imaging modalities. Our method achieves competitive results, with particularly strong performance on Microscopy (DSC Final: 0.7531, NSD Final: 0.9606), PET (DSC Final: 0.8186), and Ultrasound (DSC Final: 0.8946) modalities, where we outperform all baseline methods.

Our method demonstrates modality-specific strengths:

- **Strong Performance**: Microscopy, PET, and Ultrasound benefit from our dynamic interaction simulation, achieving best-in-class results with improvements of up to 26.7% in DSC over the next best method (Microscopy).
- **Moderate Performance**: CT shows competitive results (DSC Final: 0.8197), slightly below nnInteractive but surpassing other foundation models.

– **Weak Performance**: MRI exhibits degraded performance (DSC Final: 0.4222), significantly underperforming all baselines.

**Impact of Bounding Box Availability** A critical factor in our performance is the availability of initial bounding boxes. As shown in Table 4, our method exhibits an 81.9% drop in DSC when bounding boxes are absent, revealing a fundamental limitation of our central-third heuristic.

This dependency explains our modality-specific performance patterns. MRI suffers most severely as approximately half of its cases lack bounding boxes, while Microscopy, PET, and Ultrasound, which always provide valid bounding boxes, achieve best-in-class results. The large gap with and without bounding boxes for CT and MRI modalities underscores that our dynamic interaction simulation performs best when initialized via a bounding box.

Table 4: Performance breakdown by modality and bounding box availability

| Modality | Bbox | DSC Final | NSD Final | DSC AUC | NSD AUC |
|---|---|---|---|---|---|
| CT | No | 0.0626 | 0.0514 | 0.1850 | 0.1443 |
| | Yes | 0.8405 | 0.8731 | 3.3032 | 3.4147 |
| MRI | No | 0.1535 | 0.0811 | 0.4882 | 0.2161 |
| | Yes | 0.7987 | 0.9077 | 3.1438 | 3.5708 |
| Microscopy | Yes | 0.7532 | 0.9607 | 2.9476 | 3.8110 |
| PET | Yes | 0.8186 | 0.7841 | 3.2278 | 3.0728 |
| Ultrasound | Yes | 0.8947 | 0.9165 | 3.5364 | 3.6197 |
| **Overall with bbox** | | **0.8276** | **0.8863** | **3.2550** | **3.4774** |
| **Overall without bbox** | | **0.1499** | **0.0799** | **0.4759** | **0.2132** |

## 4.2   Qualitative results on validation set

**Analysis of Success Cases** Figures 4–8 illustrate the best and worst performing cases across all modalities, providing visual insights into our method's behavior.

Our method excels when:

– **Valid bounding boxes are provided**: The model effectively leverages spatial priors from accurate initial localization.
– **Objects have clear boundaries**: Microscopy and Ultrasound images often contain well-defined structures that benefit from our click-based refinement.

**Failure Case Analysis** The primary failure mode occurs with missing bounding boxes. Our heuristic of generating a default box at the central third of the image proves inadequate for:

- **Distributed structures**: Vessels and myocardium that span the entire volume
- **Peripheral objects**: Structures located far from the image center
- **Multiple instances**: Cases where the target object is not the central one

This limitation is most evident in MRI results, where many anatomical structures lack meaningful bounding box initialization.

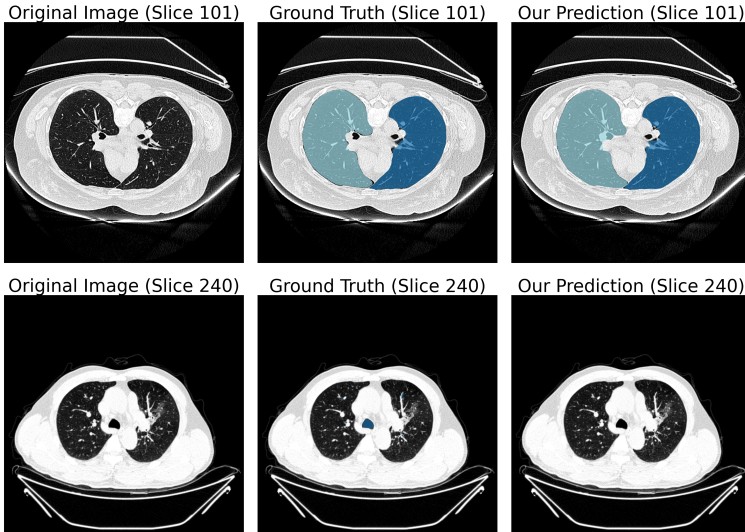

Fig. 4: Best (top) and worst (bottom) final DSC for the CT modality (`CT_LungMasks_lung_059` : final DSC = 0.9804, `CT_AirwayTree_ATM_005` : final DSC = 0)

### 4.3   Results on final testing set

Table 5 presents the performance of the method in the test set, where it ranks second among all participants.

### 4.4   Limitation and Future Work

Our primary limitation is the dependency on bounding box initialization. While our method achieves strong performance when properly initialized, it fails for anatomical structures that lack natural bounding regionsparticularly distributed structures like vessels and airways. The central-third heuristic we employ as a fallback proves inadequate, preventing our method from handling approximately half of MRI cases where bounding boxes are unavailable.

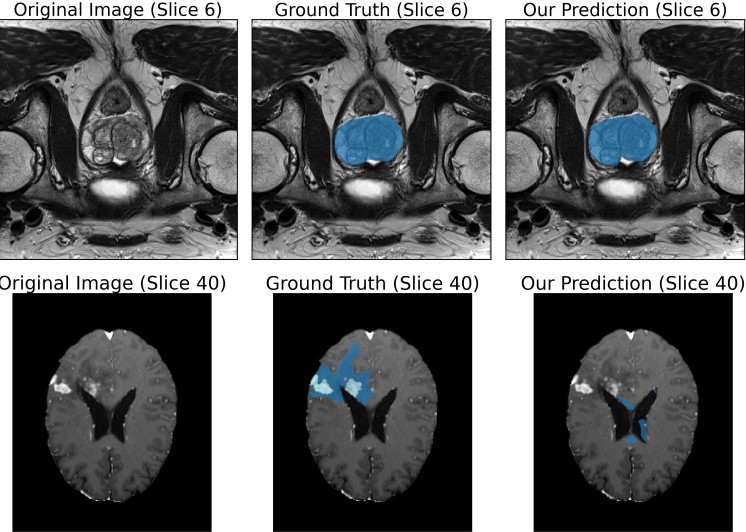

Fig. 5: Best (top) and worst (bottom) final DSC for the MRI modality (`MR_ProstateT2_T2_ProstateX_0230_4` : final DSC = 0.9475, `MR_BraTS_T1c_bratsgli_0089` : final DSC = 0.0004)

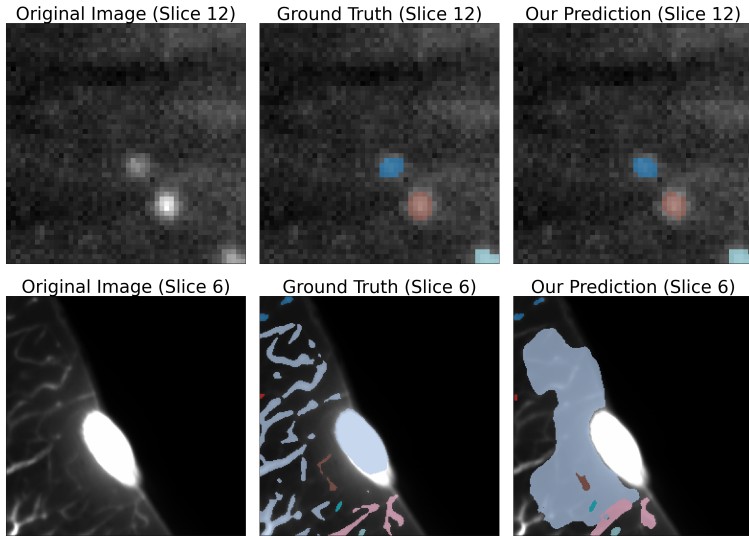

Fig. 6: Best (top) and worst (bottom) final DSC for the Microscopy modality (`Microscopy_SELMA3D_patchvolume_cfos_009_3_1_1` : final DSC = 0.9130, `Microscopy_SELMA3D_patchvolume_vessel_006_0_1_0` : final DSC = 0.5725)

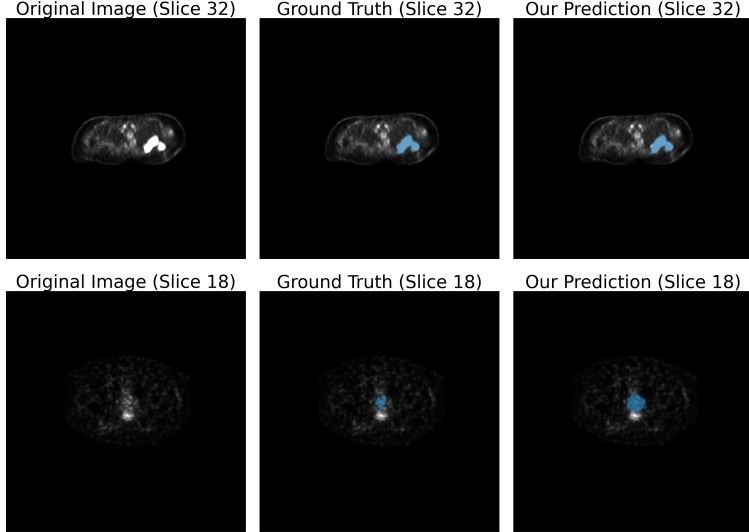

Fig. 7: Best (top) and worst (bottom) final DSC for the PET modality (`PET_autoPET_fdg_68f73c4518_11-13-2004-NA-PET-CT` : final DSC = 0.9466, `PET_autoPET_psma_a3c5675abaec3e1d_2017-08-14` : final DSC = 0.5783)

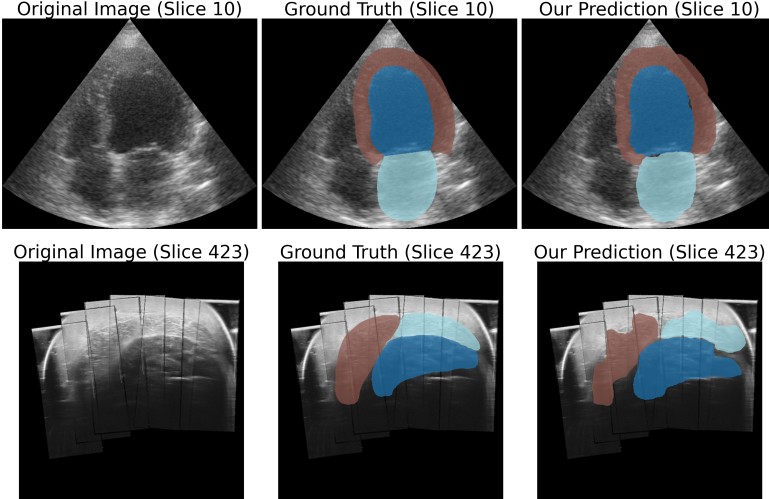

Fig. 8: Best (top) and worst (bottom) final DSC for the Ultrasound modality (`US_Cardiac_patient0224_4CH_half_sequence` : final DSC = 0.9320, `US_Low-limb-Leg40` : final DSC = 0.7316)

Table 5: Results on the test set for the **coreset track**

| Rank | Team | DSC_AUC | NSD_AUC | DSC_Final | NSD_Final |
|---|---|---|---|---|---|
| 1 | aim_coreset | 3.104 | 3.232 | 0.801 | 0.838 |
| **2** | **norateam_coreset** | **2.911** | **2.970** | **0.754** | **0.775** |
| 3 | yiooo_coreset | 2.896 | 2.898 | 0.745 | 0.749 |
| 4 | lexor_coreset | 2.501 | 2.572 | 0.625 | 0.643 |
| 5 | ahus_coreset | 2.404 | 2.266 | 0.627 | 0.597 |
| 6 | hanglok_coreset | 2.120 | 2.012 | 0.553 | 0.533 |
| 7 | cemrg_coreset | 1.885 | 1.624 | 0.476 | 0.410 |
| 8 | sail_coreset | 1.654 | 1.593 | 0.413 | 0.398 |
| 9 | dtftech_coreset | 1.591 | 1.079 | 0.417 | 0.284 |
| 10 | owwwen_coreset | 0.956 | 0.496 | 0.239 | 0.124 |

This limitation points to critical areas for future development. First, we need automated initialization strategies that can identify regions of interest without manual intervention, potentially through attention mechanisms or saliency detection. Second, the model should be trained to handle click-only interactions from the start, eliminating the bounding box dependency entirely.

## 5   Conclusion

We presented a dynamic prompt generation strategy for interactive 3D medical image segmentation that simulates realistic user interactions during training. Our method achieves state-of-the-art performance on well-initialized cases, outperforming existing foundation models on Microscopy, PET, and Ultrasound modalities with final Dice scores exceeding 0.75. However, performance critically depends on valid bounding box initialization, with an 81.9% drop in accuracy when bounding boxes are absent. This limitation highlights the need for robust initialization strategies in interactive segmentation systems. Despite this constraint, our approach demonstrates that incorporating iterative refinement patterns during training significantly improves segmentation quality when proper spatial priors are available.

**Acknowledgements**   We thank all the data owners for making the medical images publicly available and CodaLab [14] for hosting the challenge platform.

**Disclosure of Interests.**   The authors have no competing interests to declare that are relevant to the content of this article.

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

Table 6: Checklist Table. Please fill out this checklist table in the answer column. (**Delete this Table in the camera-ready submission**)

| Requirements | Answer |
|---|---|
| A meaningful title | Yes |
| The number of authors (≤6) | 3 |
| Author affiliations and ORCID | Yes |
| Corresponding author email is presented | Yes |
| Validation scores are presented in the abstract | Yes |
| Introduction includes at least three parts: background, related work, and motivation | Yes |
| A pipeline/network figure is provided | Figure 1 |
| Pre-processing | Page 7 |
| Strategies to data augmentation | Page 9 |
| Strategies to improve model inference | Page 7 |
| Post-processing | Page 7 |
| Environment setting table is provided | Table 1 |
| Training protocol table is provided | Table 2 |
| Ablation study | Page 11 |
| Efficiency evaluation results are provided | Table 3 |
| Visualized segmentation example is provided | Figures 4-8 |
| Limitation and future work are presented | Yes |
| Reference format is consistent. | Yes |
| Main text >= 8 pages (not include references and appendix) | Yes |