# OpenReview forum: "Dynamic Prompt Generation for Interactive 3D Medical Image Segmentation Training"
_thecvf.com/CVPR/2025/Workshop/MedSegFM — CVPR 2025 Workshop MedSegFM Submission_

### Official Review · Reviewer_SnJD · 2025-09-08
**Competent Use of nnInteractive for 3D Segmentation, Contribution Unclear**

**Rating:** 6
**Confidence:** 5

**Review:**

# Review

The authors propose a 3D interactive segmentation model building on nnInteractive.

## Strengths
- Good explanation of prompt encoding and model architecture, with informative figures.
- Transparent reporting of results, including failure cases when the bounding box is missing.

## Weaknesses
- It is unclear what is introduced in this work that is not already present in the nnInteractive model. The authors claim to introduce a dynamic prompt generation strategy, but from my understanding, it is the same as is used in model evaluation in the challenge.

## Novelty and Significance
The paper utilizes nnInteractive, and I fail to see any relevant novel contributions. However, novelty was not a requirement in this challenge.

## Clarity
The work is generally presented clearly, but several details should be improved in the presentation (see detailed comments).

## Suggestions for Improvement and Detailed Comments
- It might be worth mentioning that bounding boxes are optional, already in the introduction.
- Remove deep supervision from Fig. 1 instead of writing that it is not used — it is just confusing.
- In Fig. 3, you indicate that the U-Net weights are frozen — do you not update the network weights? It is unclear what is actually trained here.
- In general, equations should be treated as parts of sentences, that is, use either a dot or a comma depending on whether the sentence ends or not.
- In Eq. 13, you define your loss as "CE - 10*Dice" but in Table 2 you write "CE+Dice" — ensure you specify the correct loss function.
- For the figures in the results section, it would be helpful if the background class were transparent, and the font size were increased to roughly match the figure caption text. I also believe that it is possible to fit more than one figure per page.
- I fail to see why the proposed future work is relevant. Try motivating this a bit more.

---

> ### Author Rebuttal · Authors · 2025-10-30
>
> We thank the reviewer for their evaluation and suggestions for improving the presentation.
>
> **Unclear novelty compared to nnInteractive**
>
> We appreciate this concern and acknowledge that our work builds upon nnInteractive. However, while the challenge evaluation uses error-based click generation, nnInteractive's training strategy is neither fully documented nor publicly available. Their paper states they use "prediction error" to guide training interactions but provides no algorithmic details. We contribute a complete, reproducible implementation, which we have now clarified following reviewer feedback.
>
> We agree that novelty was not a requirement for this challenge, and our contribution lies in providing explicit, reproducible methods.
>
> **Suggestions for Improvement**
>
> We thank the reviewer for the detailed presentation suggestions and have addressed them as follows:
> - Bounding box optional in introduction: We have added this clarification in Section 1.1 (Background).
> - Deep supervision in Fig. 1: Removed from the figure to avoid confusion, as we do not use it.
> - Fig. 3 frozen weights: We apologize for this confusing notation. The U-Net weights are not frozen, we simply disable gradient computation during the first forward pass for computational efficiency. We have clarified the caption of Figure 3. We also clarified this in Section 2.3.
> - Equation punctuation: We have reviewed all equations and added appropriate punctuation.
> - Equation 13 vs Table 2 consistency : We have corrected Table 2 to include the exact loss coefficients presented in Equation (13). We confirm that the loss is Lseg=−10*LDice+LCE.
> - Results figures: The revised version includes: (1) transparent backgrounds, (2) increased font size to match caption text, and (3) multiple figures per page where appropriate.
> - Future Work Motivation : We appreciate the reviewer pointing this out. Our proposed future work directly addresses the main limitation identified in our analysis: the strong dependency on bounding box initialization (Table 4 shows 0.83 DSC with boxes vs. 0.15 without). The suggested improvements would enable the method to handle the ~50% of cases (particularly MRI) where bounding boxes are unavailable, making it applicable across all anatomical structures rather than only well-initialized cases.

---

### Official Review · Reviewer_wUDu · 2025-09-12

**Rating:** 6
**Confidence:** 4

**Review:**

This paper proposes a 3D interactive segmentation method based on nnInteractive, adding (1) a dynamic prompt generation scheme and (2) content-aware dynamic cropping.

### Strengths
- Clear explanation of prompt encoding and the network, with useful figures.
- Transparent reporting of limitations and failure cases.
- The content-aware dynamic cropping is concrete and well specified, making the engineering contribution easy to reproduce.

### Weaknesses
- Limited innovation.
- Inconsistency in the loss function.

### Suggestions for Improvement and Detailed Comments
- In Figure 3, is the U-Net completely frozen? Please clarify this in both the main text and the caption.
- Cross-check Table 2 against Equation (13) to ensure consistency.
- The font size in the figures within the experiments section is too small.

---

> ### Author Rebuttal · Authors · 2025-10-30
>
> We thank the reviewer for their careful reading and constructive feedback.
>
>
> - Figure 3 clarification
>
> Thank you for catching this ambiguity. The U-Net is not frozen during the interaction simulation stage, we simply disable gradient computation for computational efficiency. We have clarified this in the caption of Figure 3.
>
> - Loss specification
>
> We have corrected Table 2 to include the exact loss coefficients presented in Equation (13). We confirm that the loss is Lseg=−10*LDice+LCE.
>
> - Font size in experiment figures
>
> We increased the font size in Figures 4-8 to improve readability.

---

### Official Review · Reviewer_jMJt · 2025-09-14

**Rating:** 7
**Confidence:** 4

**Review:**

The paper presents a strategy to improve volumetric awareness and interactive capabilities that achieves a competitive Dice and NSD score. I have the following questions:

The paper claims `Interaction Simulation` is a key point for learning from user feedbacks. I am not sure about the statement. Should not the interactive actions are part of the training set, then why and how more simulations are generated? How the generated sets integrated to the original dataset? I would be very interested to see the ablation study on the design.

You claimed you have presented a `dynamic prompt generation` strategy. Is the prompt generation correspond to the interaction simulation part? Your claim on the method is dynamic because it is generated from a Bernoulli?

Is the, from your perspective, `Interaction Simulation` is the reason to the improvement in Table 3?

---

> ### Author Rebuttal · Authors · 2025-10-30
>
> We appreciate the positive feedback and questions from the reviewer.
>
> Q1: How are simulated interactions generated and integrated into training?
>
> We have clarified the training process in Section 2.3 (Interaction Simulation). The interaction simulation is a dynamic process occurring during training, not a preprocessing step. For each training batch:
> - The model generates an initial prediction using the image and bounding box from the dataset, without gradient computation.
> - Clicks are computed on-the-fly by analyzing errors between this prediction and the ground truth using connected component analysis.
> - A second forward pass uses all prompts (bounding box, generated clicks, and previous segmentation) with gradient computation to update weights.
> Crucially, these simulated prompts are never stored or added to the dataset; they are regenerated for every training iteration based on the current model's behavior, establishing a curriculum where the model learns from its own mistakes.
>
> Q2: What makes the prompt generation "dynamic"?
>
> "Dynamic" signifies that clicks are generated based on the current model's predictions, rather than being fixed or pre-computed. While Bernoulli sampling determines the number of clicks (0 or 1), the location and type (positive/negative) of each click adapt to the model's current errors. This creates a feedback loop where training prompts evolve as the model improves.
>
> Q3: Is Interaction Simulation responsible for the improvements in Table 3?
>
> Table 3 compares our method against other approaches, making it challenging to isolate the sole effect of interaction simulation. Notably, nnInteractive also employs prediction-dependent prompt generation during training, although their exact implementation differs and their code is not public. A controlled ablation study comparing dynamic versus static click generation would be valuable future work to precisely quantify this contribution.

---

### Author Rebuttal · Authors · 2025-10-30

We thank all reviewers for their constructive feedback. We have revised the paper to address the concerns raised:

- Clarified the interaction simulation process in Section 2.3 (R1, R3)
- Corrected Figure 1 and Figure 3 to remove confusing notations (R2, R3)
- Improved figure readability with larger fonts and better layout (R2, R3)
- Added punctuation to all equations (R3)

The updated manuscript is attached.

---

### Decision · Program_Chairs · 2025-11-12

Accept